# A Mantle Plume Beneath South China Revealed by Electrical Conductivity Obtained from Three-Dimensional Inversion of Geomagnetic Data

**DOI:** 10.3390/s23031249

**Published:** 2023-01-21

**Authors:** Shiwen Li, Yunhe Liu, Jianping Li

**Affiliations:** 1College of Geo-Exploration Science and Technology, Jilin University, Changchun 130026, China; 2Key Laboratory of Marine Mineral Resources, Ministry of Natural Resources, Guangzhou Marine Geological Survey, China Geological Survey, Guangzhou 510075, China

**Keywords:** earth observation, geomagnetic data, three-dimensional imaging, Hainan plume, geomagnetic depth sounding

## Abstract

A three-dimensional electrical conductivity model of the mantle beneath South China is presented using the geomagnetic depth sounding method in this paper. The data misfit term in the inversion function is measured by the L1-norm to suppress the instability caused by large noises contained in the observed data. To properly correct the ocean effect in responses at coastal observatories, a high-resolution (1° × 1°) heterogeneous and fixed shell is included in inversion. The most striking feature of the obtained model is a continuous high-conductivity anomaly that is centered on ~(112° E, 27° N) in the mantle. The average conductivity of the anomaly appears to be two to four times higher than that of the global average models at the most sensitive depths (410–900 km) of geomagnetic depth sounding. Further analysis combining laboratory-measured conductivity models with the observed conductivity model shows that the anomaly implies excess temperature in the mantle. This suggests the existence of a mantle plume, corresponding to the Hainan plume, that originates in the lower mantle, passes through the mantle transition zone, and enters the upper mantle. Our electrical conductivity model provides convincing evidence for the mantle plume beneath South China.

## 1. Introduction

Mantle plumes can transport material and energy to the surface. They may be closely related to the breakup of the paleo-continents, the formation of large igneous provinces, the eruption of a series of intraplate volcanoes, and even mass extinctions [1,2]. Therefore, the reconstruction of mantle plumes plays a key role in understanding mantle convection and Earth’s evolution. The temperature and mineral compositions of a plume are obviously different from those of the surrounding mantle, and a mantle plume interacts with the mantle transition zone (MTZ) and other interfaces in Earth during its rise, causing significant changes in the structure of the mantle [3]. Therefore, the most convincing evidence for the existence of a mantle plume comes from geophysical observations of the mantle.

Geophysical methods, particularly seismology and seismic tomography, are used to detect plumes. Many achievements have been made in plume research through seismic imaging [4,5,6,7,8]. Recently, a low-velocity structure was discovered in the mantle below Hainan Island [9,10,11,12,13,14]; this structure originates in the lower mantle, crosses the MTZ, and reaches the lithosphere. This characteristic suggests that the low-velocity structure is a mantle plume. Therefore, the Hainan plume has been put forward to explain the formation and evolution of the South China Sea, the mechanism of Hainan’s volcanoes, and other scientific issues. However, there are still large disputes among different research groups about the location, shape, temperature, and other properties of this mantle plume. For example, Huang [12] found that the tail of the Hainan mantle plume is located northeast of the Hainan volcanoes through seismic P-wave and S-wave imaging, with a diameter of approximately 200 km, whereas Xia et al. [10] found the tail of the Hainan mantle plume to be more northerly, located at the junction of Guangdong, Guangxi, and Hunan Provinces, with a diameter of 200~300 km, via imaging using the teleseismic method. Such inconsistencies make discussion of the Hainan plume difficult. The application of other geophysical methods will help to reach a consensus.

The electrical conductivity of the mantle is sensitive to variations induced by plumes. Therefore, geomagnetic depth sounding (GDS), which can reveal the conductivity of the mantle [15], has become an important method of plume detection. Although the resolution of GDS is slightly lower than the diameter of a plume tail, when a plume impinges on the MTZ, the exothermic reaction caused by the phase transformation at the 660 km interface can result in the accumulation of a tremendous volume of hot plume materials beneath the MTZ [16]. The extent of the head formed by these materials may exceed 1000 km in diameter [3]. This accumulation significantly increases the conductivity near the MTZ and makes it possible to detect the existence of mantle plumes via GDS. The distribution of geomagnetic stations in South China and its surrounding areas is relatively dense, providing unique data conditions for the application of GDS to detect the Hainan mantle plume.

## 2. Data and Methods

### 2.1. GDS Theory

The inducing source for GDS is the slowly changing ring currents in the magnetosphere [17]. These currents are concentric with the magnetic equator of Earth, and so the numerical simulation is developed based on the geomagnetic spherical coordinate system. The widely used C-response of GDS is estimated from the $H_{r}$ (the vertical component pointing downwards towards the center of Earth) and $H_{\theta }$ components (the colatitudinal component pointing towards magnetic north) of the magnetic field (**H**) at the surface. With the assumption that the ring currents can be described by the spherical harmonic function $P_{1}^{0}$ [17,18,19], the *C*-response can be calculated by
(1)$$C\left(\omega \right)=-\frac{a_{0}\tan\theta }{2}\frac{H_{r}\left(\omega \right)}{H_{\theta }\left(\omega \right)},$$
where $a_{0}$ is the average radius of Earth, ω is the angular frequency, and *θ* represents the geomagnetic colatitude (0–180°). The induced geomagnetic signal collected at the surface for GDS has a period of several days to more than 100 days.

Equation (1) shows that C-responses should be calculated from the magnetic field **H**. Under the assumption of positive time harmonic dependence of the form $e^{i\omega t}$, **H** obeys
(2)$$\nabla \times \left(\rho \nabla \times H\right)+i\omega \mu_{0}H=0,$$
where ρ is the reciprocal of electrical conductivity σ, $\mu_{0}$ is the vacuum magnetic permeability, and i is the imaginary unit. Equation (2) can be solved by means of the staggered-grid finite difference method in a spherical coordinate system [20]. The model parameterized for calculation includes resistive air and conductive Earth. The outer boundary of air is 2$a_{0}$ from the surface, and its resistivity is set to a moderately large finite value of 10^10^ Ω·m. The inner boundary of Earth is the core–mantle boundary (CMB) due to the superconductive core [17]. The tangential components of **H** at the boundaries are specified so that Equation (2) holds throughout the space domain, while the resultant numerical system remains acceptably well-conditioned. The location of the $P_{1}^{0}$ source is placed at a radial distance from Earth’s surface of 10$a_{0}$ to ensure that the secondary magnetic field induced by the conductive Earth can be considered negligible. A variant of the biconjugate gradient and an iteration method are used to obtain the solution of discretized Equation (2) [21]. To ensure that **H** is conservative during iteration, the divergence correction [22] is also applied.

### 2.2. L-BFGS Inversion

GDS inversion can generally be expressed as an optimization problem
(3)$$\Phi \left(m,\lambda \right)\underset{m}{\to }\min,$$
where the penalty function $\Phi \left(m,\lambda \right)$ is defined by
(4)$$\Phi \left(m,\lambda \right)=\Phi_{d}\left(m\right)+\lambda \Phi_{m}\left(m\right),$$
where $\Phi_{d}\left(m\right)$ and $\Phi_{m}\left(m\right)$ are data misfit and model roughness, respectively, and λ is the regularization parameter, which is used to trade off $\Phi_{d}\left(m\right)$ and $\Phi_{m}\left(m\right)$. m is the electrical conductivity vector, and in the case of three-dimensional (3D) inversion, it represents the conductivity in each prism [5].

Using the notation of *L*_p_-norm measurement of the objective function, Equation (4) is expressed by
(5)$$\Phi \left(m,\lambda \right)=\|W_{d}{\left(\psi \left(m\right)-d\right)\|}_{p}^{p}+\lambda \|W_{m}{\left(m-m_{0}\right)\|}_{p}^{p},$$
where **d** is the data vector, $m_{0}$ is the prior model, ψ represents the forward mapping operator used to calculate the responses of model m, $W_{d}$ is a diagonal matrix with data covariance as diagonal elements, and $W_{m}$ is a smoothing matrix used to relate the conductivity of each grid to that of the adjacent grids in three directions (X, Y, and Z directions). The correlation among adjacent cells increases as the value of the smoothing coefficients increases from 0 to 1. *L*_p_-norm inversion can be realized by assigning different values of p.

In the traditional inversion approach, differentiating both sides of Equation (5) with respect to the space domain model parameters and neglecting the higher order terms of the Taylor-series expansion allow the linear system of equations to be solved at each iteration, as follows:(6)$$\left[J^{T}W_{d}^{T}R_{d}W_{d}J+\lambda W_{m}^{T}R_{m}W_{m}\right]\delta m=J^{T}W_{d}^{T}R_{d}W_{d}\left[\psi \left(m\right)-d\right]+\lambda W_{m}^{T}R_{m}W_{m}\left(m-m_{0}\right),$$
where
(7)$$R_{j}\left(x\right)=p{\left(x^{2}+\varepsilon^{2}\right)}^{p/2-1}\begin{array}{cc} , & j=\begin{array}{ccc} d & or & m \end{array} \end{array},$$
where *ε* is a small number to ensure a solution when x=0 and p corresponds to the *L*_p_-norm inversion.

The optimization of Equation (3) in the case of GDS inversion is nonlinear, and we select the limited-memory quasi-Newton method (L-BFGS), which has been widely used in electromagnetic induction exploration [23] to seek the solution of the penalty function. L-BFGS is a modified form of the quasi-Newton method. The basic iteration formula of L-BFGS is
(8)$$m_{k+1}=m_{k}+\alpha_{k}p_{k},$$
where
(9)$$p_{k}=-B_{k}^{-1}\nabla \Phi_{k},$$
and
(10)$$\nabla \Phi_{k}={\left.{\left(\frac{\partial \Phi }{\partial m_{1}},\frac{\partial \Phi }{\partial m_{2}},\cdots ,\frac{\partial \Phi }{\partial m_{N}}\right)}^{T}\right|}_{m=m_{k}},$$
where k is the number of iterations, $\alpha_{k}$ is the searching step, $p_{k}$ is the searching direction, and $B_{k}$ is the approximation of the Hessian matrix [24]. The approximation of the Hessian matrix avoids calculating the Hessian matrix directly, thus tremendously reducing the requirements for computer storage and computation time.

The computation of Equation (6) requires the calculation of the Jacobian matrix and forward responses. The latter can be easily accomplished. Nominally, the Jacobian matrix can be directly computed, but a more feasible method, the adjoint forward technique, can be considered [21,22,25]. Using this method, we compute the product of the matrix and the data vector, which is split into a few adjacent forward operations, thus greatly reducing computational requirements.

### 2.3. Data Processing

C-responses can be estimated using the BIRRP software package [26], which is a combination of the standard M estimation method and hat matrix diagonal statistical analysis; thus, it can eliminate the interference in $H_{r}$ and $H_{\theta }$ and correlated noise in both. The application of the remote reference method is suggested in BIRRP, but the self-reference method has been proven to show only a negligible difference from the remote reference method [27] in the long periods used for the present work; therefore, in this paper, we estimate C-responses by means of the self-reference method based on BIRRP.

Hourly mean value time series of the three components of the geomagnetic field can be obtained from the World Data Center. In this paper, the selected time series are from 6 years up to approximately 60 years. In addition, the Geomagnetic Network of China provides data from deployed geomagnetic stations in China, which significantly enhances the number of stations that can be used in GDS inversion. After careful selection according to the duration and noise that we presented previously [21], we obtained 15 geomagnetic observatories in South China for further consideration (Figure 1). The details of the observatories are listed in Table 1. The estimated C-responses and their errors within 16 periods from 3.5 to 113 days at the selected stations are presented in Figure 2. The estimation of C-responses in the frequency domain strongly depends on the data length in the time domain. However, the data length varies at different stations, besides the observed data being strongly affected by environmental noise and the ocean effect, leading to a huge variability both in the real and imaginary parts of C-responses, especially for responses in a long period. The variation in responses at different stations indicates that the electrical structure beneath South China is heterogeneous. The squared coherency of C-responses, which is commonly treated as a quality indicator of C-responses, is depicted in Figure 2c, showing that most of the C-responses are of good quality, while some stations produce C-responses of poor quality. Considering the strong fluctuation for some stations, the *L*_1_-norm measurement is used to measure the data misfit to suppress the influence of data noise on the inversion results [21].

### 2.4. Influence of the Ocean

The C-responses at coastal observatories are significantly influenced by ocean induction effects (OIEs) due to the large contrast in conductivity between oceans and continents, especially for responses over short periods [27,28,29,30]. Some of our selected observatories are located near the coastline, so we examine the OIEs on C-responses. Based on the global average one-dimensional (1D) model (Figure 3) derived from vector magnetic field data observed by satellite [31], we calculated the C-response at the Guangzhou (GZH) and Sheshan (SSH) stations near the coastline. The C-responses of the 1D model covered by a shell comprising oceans and continents at the two stations were also calculated. As shown in Figure 4, both the real and imaginary components of the C-response are significantly affected in the examined periods. The influence on the imaginary components is more obvious, and the OIEs are stronger in GZH than in SSH. The influence of the OIEs at the two stations can reach a period of 30 days if 5% variations are taken as a reference, which are commonly used as an error floor in electromagnetic induction inversion [32,33,34]. According to the relationship between electrical conductivity and the C-response, the conductivity is proportional to the imaginary component of the C-response, and the penetration depth is directly proportional to the real component. The strong variations in C-responses correspond to the extremely enhanced conductivity in the shallow mantle, which do not coincide with the actual situation. Therefore, the response of stations affected by OIEs must be corrected.

The ratio method can be used for OIE correction [35], and the specific correction formula is as follows:(11)$$C^{\mathrm{corr}}=C^{o\mathrm{bs}}\cdot k,\;\mathrm{and}\;k=C^{1D}/C^{1D+\mathrm{shell}}$$
where $C^{\mathrm{corr}}$ is the corrected response, $C^{o\mathrm{bs}}$ is the observed response at the station, and k is the correction coefficient, which can be calculated from the response of the 1D model and the response of the 1D model with a covering shell comprising oceans and continents. Obviously, the correction coefficient changes with the 1D model. This makes it difficult to identify a 1D model that can be suitable for all stations because the electrical structure of Earth is unknown and three-dimensional. Therefore, in our 3D inversion of GDS, the conductivity of the shell with ocean and land was treated as the surface layer of Earth. The shell was considered in the forward numerical modeling but was fixed throughout the inversion [21,36].

## 3. Inversion Results and Stability

### 3.1. Electrical Conductivity Model

The C-responses (Figure 2) at the 15 stations in South China were inverted via 3D GDS inversion, in which the data misfit and model roughness were measured using *L*_1_- and *L*_2_-norm measurements, respectively. To reduce the influence of the electrical structures of the surrounding mantle, stations near the research area were also considered in our inversion. The data error used to normalize the data misfit in inversion was obtained from the estimated responses. Data with large uncertainties should be excluded from the traditional L2-norm inversion, but in *L*_1_-norm inversion, their contributions could be suppressed by data error normalization and *L*_1_-norm measurement. Therefore, it is expected that we could obtain a reliable 3D electrical structure of the mantle beneath South China.

The initial model of our 3D *L*_1_-norm inversion adopted the 1D model in Figure 3. To match the major mineral phase transitions in the mantle, the conductivity was allowed to jump at 410 km, 520 km, and 670 km. A heterogeneous grid was densified with a fineness of 3° × 3° horizontally to discretize the model in the research area, and the size of the cell increased gradually outside the area (as shown in Figure 5). To eliminate the OIEs, a lateral grid of 1° × 1° of the surface layer with a thickness of 12.65 km was considered to more precisely describe the distribution of the sea and continent to generate the shell in 3° × 3° for calculation and ensure OIEs were accounted for with sufficient accuracy [28].

The 3D inversion started with an initial regularization parameter of 1.0. During iterations when the relative change in the data fitting error was less than the given threshold value, the regularization parameter was halved. The inversion process continued iteratively until the target data misfit was reached or the regularization parameter was rather small. The inversion was repeated for a number of different regularization parameters to better assess the stability of any detected mantle anomalies. The results of these inversions show that the distribution of anomalies is similar for different inverse models, except that the boundary sharpness is different. After 59 iterations, the inversion terminated with a regularization parameter smaller than 10^−4^. The root mean square (RMS) of data misfit is 1.81, which is larger than the expected value of 1.0. This large RMS is due to some unreliable data included in the inversion. If we exclude the responses whose RMS is larger than 5.0, the RMS decreases to 1.19. When using *L*_1_-norm inversion, the models from the two inversions share almost the same electrical structures. The results of inversions with an initial regularization parameter equaling 100 and stations excluding responses with a large RMS are shown in Figure 6 to strengthen the reliability of our preferred model. The data fitting curves are plotted in Figure 7. The curves show a good fitting for most responses, implying a convincing result for our inversion.

Figure 8 displays our preferred 3D electrical conductivity distribution in the mantle (250–1600 km) beneath South China, corresponding to the sensitive depth of GDS. The most noteworthy feature (Anomaly A) is the enhanced conductivity area, which is centered on ~ (112° E, 27° N). The anomaly is nearly vertical and continuous. It can be found in the lower mantle, MTZ, and upper mantle. The strongest variation in conductivity occurs in the lower MTZ, with conductivity reaching approximately 7 S/m at the center. However, this extremely high conductivity is not suitable for further analysis, because the electromagnetic induction is primarily sensitive to the integrated conductance of a conductivity body which prefers to generate a compensating higher conductivity in the core [37]. In the topmost lower mantle (670–900 km), the anomaly is similar to that in the lower MTZ both in shape and conductivity value. The average conductivity of Anomaly A appears to be two to four times higher than that of the global average models at the most sensitive depths (410–900 km) of GDS. The variation in conductivity is much weaker in the upper MTZ (410–520 km), but enhanced conductivity is observed in the shallow upper mantle. Although Anomaly A seems to extend to the deeper lower mantle (900–1200 km), we could not confirm this extent because these depths were outside of the sensitive zone of GDS, and the anomaly may have been caused by leakage from the conductive zone at 670–900 km [22]. Therefore, the anomaly at these depths will not be discussed here.

### 3.2. Robustness of Anomaly A

To determine whether Anomaly A is strictly required by the data, we repeated the *L*_1_-norm inversion with the same control parameters but set a hard prior bound in the region where the cell conductivity is fixed to that of the starting model and not permitted to vary during the iterative inversion as a free parameter. The results of the re-inverted test are shown in Figure 9 and Figure 10.

Figure 9 shows that if the area of Anomaly A is restricted, the inversion produces new conductive areas around the periphery of the previously conductive zone (rectangle with red dashed line), and the new areas are the most obvious at depths of 670–900 km. The conductivity of the zone from 900 to 1200 km increases to a rather large value compared to that of the preferred model. Figure 10 shows the fitting curves of C-responses for three observatories (CDP, GZH, and WHN) located near the region of Anomaly A. It is evident that when suppressing the conductive feature by setting a hard prior bound (i.e., the conductivity in this zone must be equal to the background conductivity), the model response misfit deteriorates significantly at the three stations, particularly at GZH and WHN. The response shifts considerably from the observed data at nearly all periods, meaning that the conductivity at all depths is inconsistent with the real mantle situation. We therefore conclude that Anomaly A is a required feature of the data.

## 4. Discussion

The conductivity of the mantle is affected by the mineral composition, temperature, and volatiles such as water [38,39]. The conductivity values of minerals measured under high-temperature and high-pressure conditions with varying water contents in the laboratory [40] allow us to explore the properties of Anomaly A. Heterogeneity in composition and temperature is mainly created by subducting slabs and mantle plumes, while water is carried and released by subducting slabs. The contribution of water can be excluded because there is little evidence of a slab that penetrates or stagnates in the MTZ beneath South China. This exclusion can be strengthened by the elevated 660 km discontinuity at the location measured using the receiver function technique [41], which is contradictory to the depression caused by water [42]. The influence of a plume on conductivity is more dependent on the excess temperature rather than on compositional differences. Therefore, the enhanced conductivity of Anomaly A may be a result of high temperatures.

In the lower mantle, the conductivity of the lower mantle can be estimated by
(12)$$\sigma =\sigma_{0}\exp(-\frac{\Delta E+P\Delta V}{kT}),$$
where σ is the electrical conductivity of the lower mantle, $\sigma_{0}$ is the pre-exponential factor, and k is the Boltzmann constant. P and T are the pressure and temperature, and their values at different depths in the mantle can be extracted from Xu et al. [43]. ΔE and ΔV are the activation energy and activation volume, respectively. Taken from the literature [43], $\sigma_{0}=74\;S/m$, ΔE=0.7 ev, and $\Delta V=-0.55\pm 0.01\;{\mathrm{cm}}^{3}/\mathrm{mol}.$ In the uppermost lower mantle, the average temperature is approximately 1900 K, which can be calculated from the global average conductivity according to Equation (12) and matches the result presented by Xu et al. [43]. To fit the average conductivity of Anomaly A, a temperature at approximately 2300 K, which is 400 K higher than the global average, is required (Figure 11b). Additionally, the temperature at the CMB is approximately 2550 K; assuming an adiabatic upwelling mantle plume originating from the CMB, this estimate is reasonable.

In the MTZ, wadsleyite and ringwoodite have a relatively high water capacity [44,45]. We can estimate the average water contents in the MTZ according to the global average 1D model and the electrical conductivity model of Yoshino et al. [38]
(13)$$\sigma =\sigma_{0H}\exp\left(-\frac{H_{H}}{kT}\right)+\sigma_{0P}C_{w}\exp\left(-\frac{H_{P}^{0}-\alpha C_{w}^{1/3}}{kT}\right),$$
where $C_{w}$ is the water content, $\sigma_{0}$ is the pre-exponential factor, α is a constant accounting for geometrical factors, and H is the activation enthalpy. Subscripts H and P denote small polaron and proton conduction, respectively. The values of parameters contained in Equation (13) measured by Yoshino et al. [38] are shown in Table 2. The estimated average water content in the MTZ is used to evaluate the temperature of Anomaly A. Figure 11b shows that in the lower MTZ, the conductivity of ringwoodite with a temperature approximately 300 K higher than the geotherm is close to the average conductivity of Anomaly A. In the upper MTZ, a comparable temperature approximately 200 K higher than the geotherm is obtained based on the average conductivity.

In the upper mantle, the conductivity is mainly determined by olivine. The conductivity of hydrous olivine has been measured [46] and can be calculated by
(14)$$\sigma =\sigma_{0}^{Vacancy}\exp\left(-\frac{\Delta H^{Vacancy}}{RT}\right)+\sigma_{0}^{Polaron}\exp\left(-\frac{\Delta H^{Polaron}}{RT}\right)+\sigma_{0}^{Hydrous}C_{\omega }\exp\left(-\frac{\Delta H^{Hydrous}-\alpha C_{\omega }^{1/3}}{RT}\right),$$
where $\sigma_{0}$ and ΔH are the pre-exponential factors and activation enthalpies, respectively. The superscripts vacancy, polaron, and hydrous represent the conductive mechanisms in olivine. Values of the parameters measured by Gardés et al. [46] are listed in Table 3. Anomaly A in the deep upper mantle can be explained by a temperature enhanced by approximately 200 K in the region.

Through the above analysis, Anomaly A is seen to be caused by high temperature. The temperature is highest in the lower mantle and gradually decreases in the MTZ and upper mantle. The correlation in the location of this structure in these different zones indicates that the increased temperatures have a common origin. Therefore, we speculate the existence of a mantle plume beneath South China, which is commonly defined as the Hainan plume. As shown in our electrical structure, the Hainan plume originates from the lower mantle. The low viscosity of the high-temperature mantle plume causes it to rise rapidly in the lower mantle, forming a narrow tail that is beyond the resolution of GDS. When the plume impinges on the MTZ, the exothermic reaction caused by the phase transformation at the 670 km discontinuity interface can block upwelling and result in the accumulation of a tremendous volume of hot plume materials beneath the MTZ, forming a large head [47]. Simultaneously, the bottom of the MTZ is heated by the impeded hot plume, significantly enhancing conductivity in the lower MTZ. The broad head and heated lower MTZ are revealed in GDS images, as shown in Figure 7 and Figure 11a. In the MTZ, the viscosity differences between the plume and wadsleyite and ringwoodite allow the plume to pass through the MTZ rapidly through a narrow channel. Therefore, it is difficult to detect the plume itself if the MTZ heated by the plume is ignored. This is the reason why only slight conductivity increases are observed in the upper MTZ (Figure 11a). The corresponding relationship between location and temperature between the upper mantle and the lower mantle is a clear indicator that the Hainan plume passes through the MTZ and enters the upper mantle. However, limited by the detection depth of GDS, we are unable to trace the rising plume in the upper mantle.

Our deduction is also supported by previous seismological images. Xia et al. [10] discovered a continuous low-velocity anomaly in and around the MTZ, which extends down to the lower mantle deeper than 1100 km, as marked by the black circles in Figure 7. Considering that the electrical structure not covered by geomagnetic stations cannot be well constrained by GDS, we only show the seismic structure in the region covered by our selected stations. The low-velocity anomaly is suggested to be the tail and head of the Hainan plume, which originates in the lower mantle and feeds the Hainan hotspot. Other wave velocity structures [48,49,50] and perturbations of 410 km and 660 km discontinuity [41] also suggest the existence of the Hainan plume. The high temperature associated with the Hainan plume can concurrently decrease the seismic velocity and increase the electrical conductivity significantly [3].

## 5. Conclusions

In this paper, we have presented the first 3D electrical structure of the mantle beneath South China from geomagnetic stations in the area. The *L*_1_-norm inversion method was used to suppress the influence of data with considerable noise, and model roughness was measured by an *L*_2_-norm as in the traditional inversions to obtain a smooth model. The self-reference method based on BIRRP was used to obtain the C-response data used in our inversion in periods from 3.5 to 113 days, making GDS sensitive to conductivity at depths of 250–1600 km. The inverted model revealed the presence of a vertical and continuous conductivity anomaly in the depth range of 250–900 km centered on (112° E, 27° N).

The results of the laboratory measurements of the electrical conductivity of mantle minerals at high temperatures and high pressures were used to interpret the nature of Anomaly A. This analysis strongly suggests that the anomaly is caused by the high temperature of the Hainan plume. The electrical conductivity model shows that the Hainan plume originates in the lower mantle and that it passes through the MTZ rapidly and ultimately enters the upper mantle. We provide convincing evidence for the existence of the Hainan plume based on the electrical structure of the mantle converted from the geomagnetic data in South China via 3D GDS inversion.

The detection depth of GDS limits its ability to distinguish the Hainan plume in the shallow upper mantle. Therefore, appropriate methods (e.g., the long-period magnetotelluric method) can be used to assess the electrical structure in the upper mantle. Therefore, future studies with joint inversion of GDS and long-period magnetotelluric data will strongly advance the exploration of the Hainan plume.

## Figures and Tables

**Figure 1 sensors-23-01249-f001:**
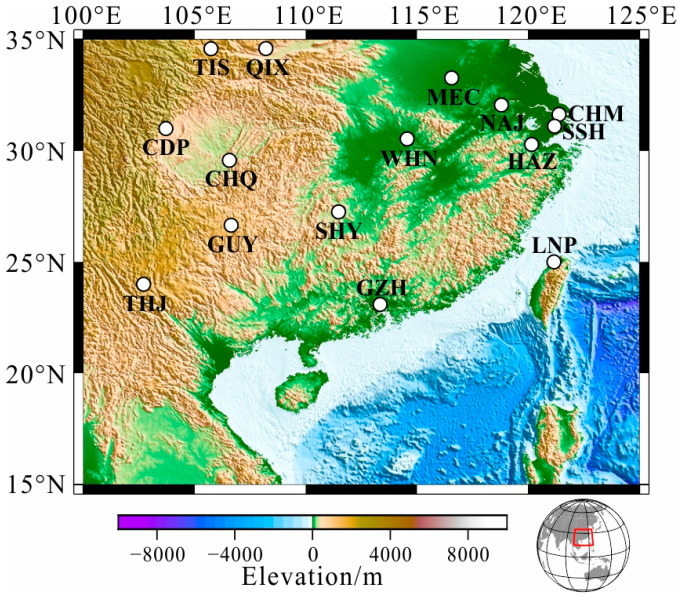
Locations of the selected geomagnetic observatories in South China.

**Figure 2 sensors-23-01249-f002:**
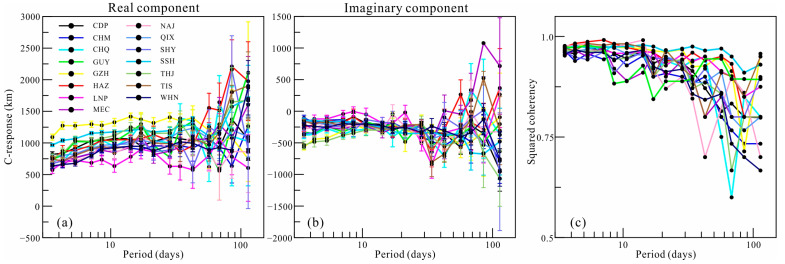
C-responses and their squared coherency at the 15 selected observatories. The lines in (**a**,**b**) are the real and imaginary components of the C-responses, and vertical lines are the data errors. (**c**) The variations in the squared coherency of C-responses.

**Figure 3 sensors-23-01249-f003:**
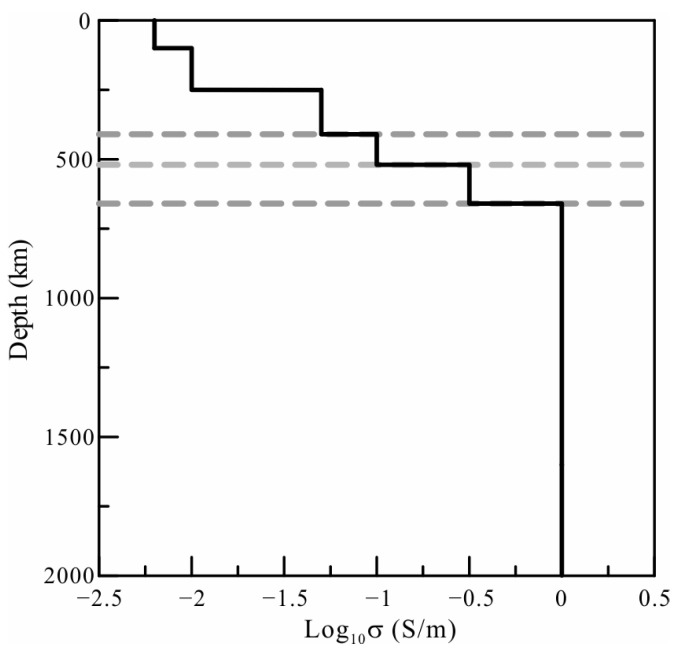
The global average one-dimensional (1D) background model used in this paper.

**Figure 4 sensors-23-01249-f004:**
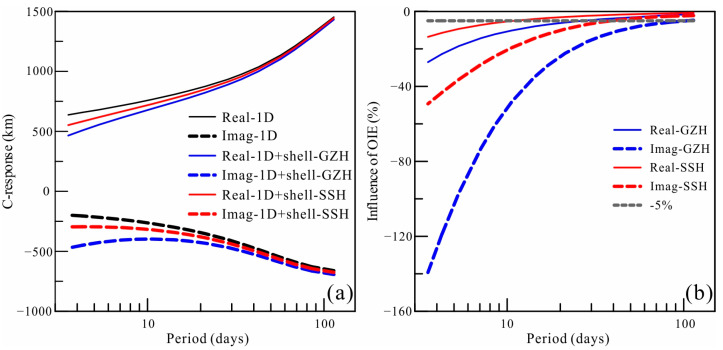
Ocean induction effects at Guangzhou (GZH) and Sheshan (SSH). (**a**) are the real and imaginary components of C-responses calculated by 1D model and 1D model with a surface shell comprised with ocean and land at GZH and SSH. (**b**) are the variations on the real and imaginary components of C-responses caused by the OIE.

**Figure 5 sensors-23-01249-f005:**
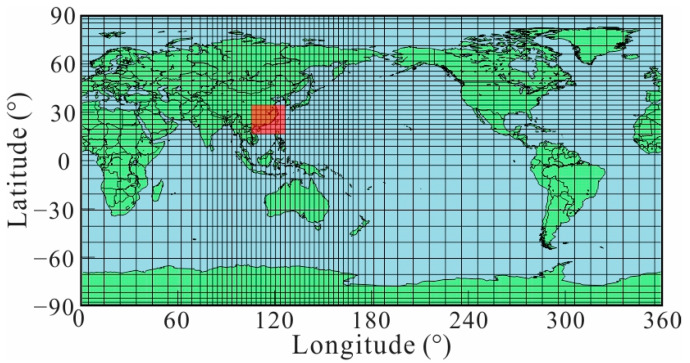
The heterogeneous grid used in our inversion. The red zone is the research area of this paper.

**Figure 6 sensors-23-01249-f006:**
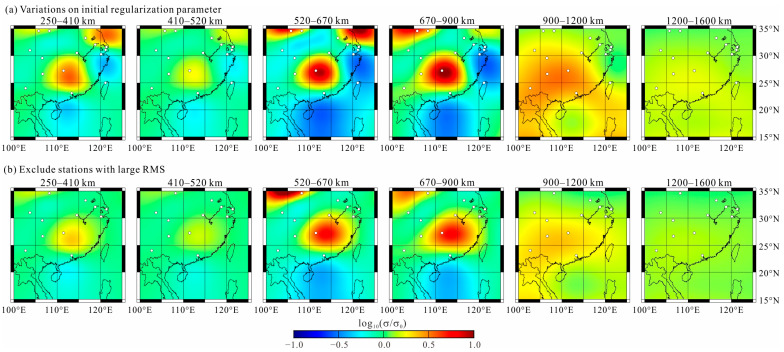
The results of inversions with an initial regularization parameter equaling 100 (**a**) and stations excluding responses with a large RMS (**b**).

**Figure 7 sensors-23-01249-f007:**
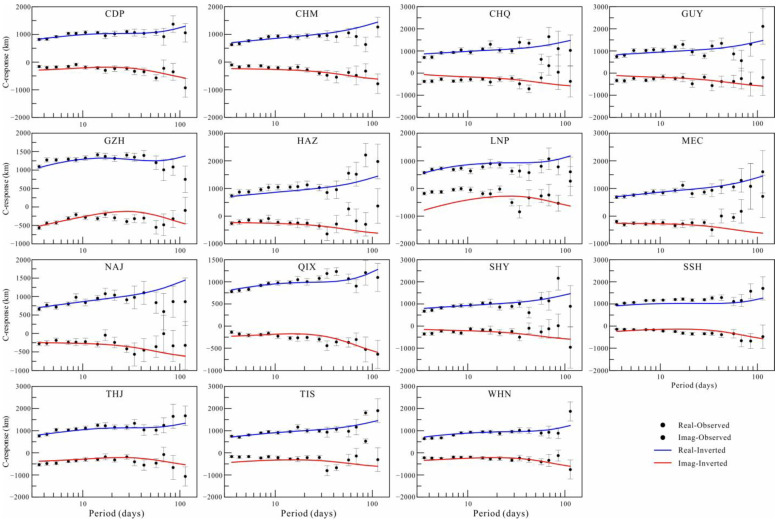
Fitting curves of the inverted responses and observed data.

**Figure 8 sensors-23-01249-f008:**
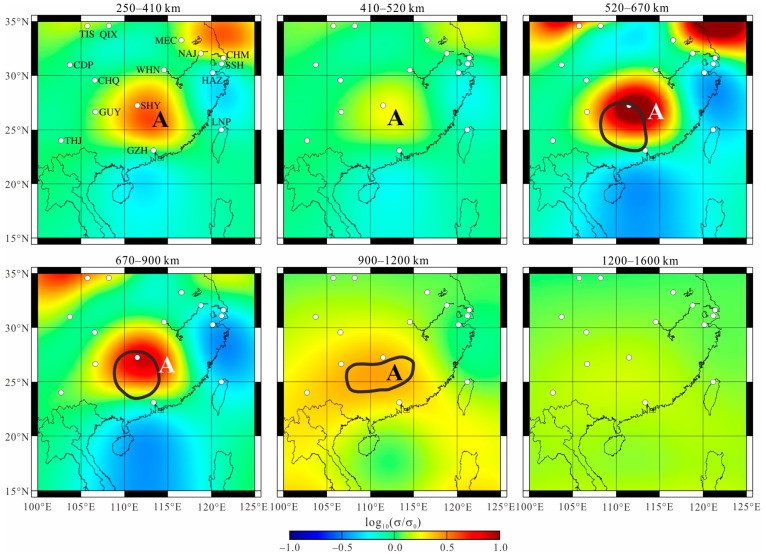
Electrical conductivity model of the mantle beneath South China. Label A represents the high conductivity zone named anomaly A. The black lines are the low-velocity zones observed via the teleseismic imaging of Xia et al. [10].

**Figure 9 sensors-23-01249-f009:**
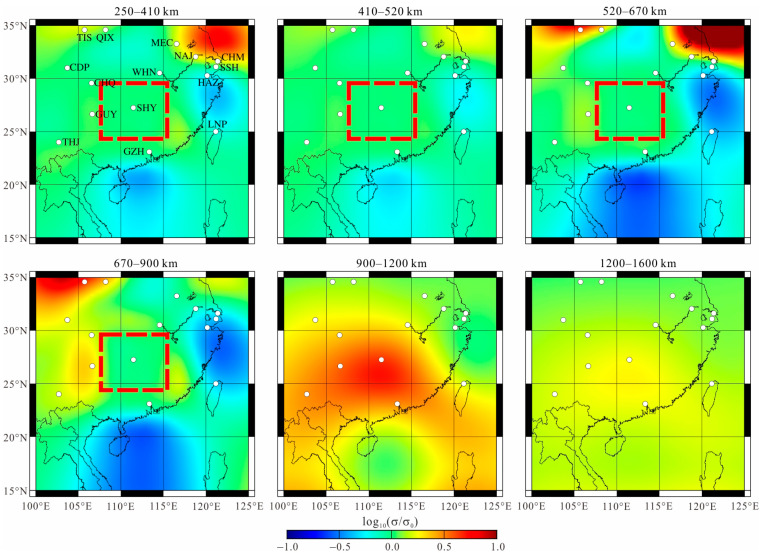
The inversion results of restricting the area (between depths of 250 km and 900 km, marked by red dotted rectangles) to be resistive and fixed during inversion.

**Figure 10 sensors-23-01249-f010:**
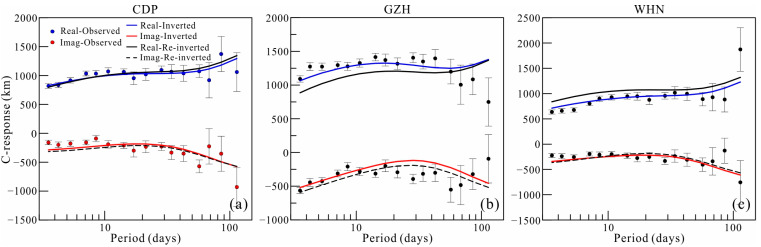
Fitness curves of the response of inversion results, re-inverted results, and the observed C-responses for three geomagnetic observatories (CDP (**a**), GZH (**b**), and WHN (**c**)) distributed around Anomaly A.

**Figure 11 sensors-23-01249-f011:**
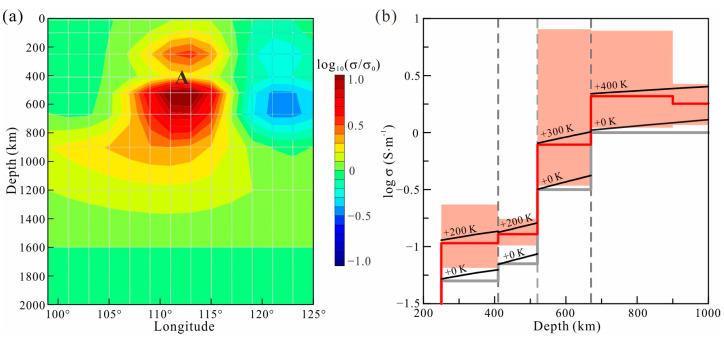
The conductivity cross-section along 27° N (**a**) and the rock physics model used to interpret the conductivity of Anomaly A (**b**). The conductivity of Anomaly A is delimited in the light red zone, with the red line denoting its average conductivity. The black lines with labeled temperatures represent the conductivity of minerals with corresponding temperatures.

**Table 1 sensors-23-01249-t001:** Station details of the observatories used in this article. GM corresponds to the geomagnetic coordinates, and the data length is the duration of the recorded geomagnetic field data time series at the station. WDC means the data were downloaded from the World Data Center, and GNC means the data were obtained from the Geomagnetic Network of China.

Code	Name	Longitude	Latitude	Gm_Long	Gm_Lat	Data Length	Data Source
CDP	Chengdu	103.7	31	176.11	20.89	1995–2017	WDC
CHM	Chongming	121.4	31.63	192.14	21.74	1995–2016	GNC
CHQ	Chongqing	106.56	29.57	178.72	19.44	1995–2001	GNC
GUY	Guiyang	106.64	26.65	178.78	16.52	1995–2013	GNC
GZH	Guangzhou	113.34	23.09	184.91	12.99	1960–2017	WDC
HAZ	Hangzhou	120.16	30.28	191.06	20.35	1995–2001	GNC
LNP	Lunping	121.17	25	192.22	15.11	1980–2000	WDC
MEC	Mengcheng	116.56	33.27	187.66	23.23	1995–2001	GNC
NAJ	Nanjing	118.8	32.06	189.74	22.09	1995–2001	GNC
QIX	Qianxian	108.2	34.6	180.05	24.46	1995–2015	WDC
SHY	Shaoyang	111.47	27.24	183.1	17.13	1995–2016	GNC
SSH	Sheshan	121.19	31.1	191.97	21.2	1932–2006	WDC
THJ	Tonghai	102.7	24	175.07	13.91	1995–2017	WDC
TIS	Tianshui	105.73	34.59	178	24.46	1995–2001	GNC
WHN	Wuhan	114.56	30.53	185.9	20.45	1995–2017	WDC

**Table 2 sensors-23-01249-t002:** Value of parameters contained in Equation (13). Numbers in parentheses are the errors by nonlinear least squares fitting (1s standard deviation).

Mineral	$\sigma_{0H}\;\left(S/m\right)$	$H_{H}\;\left(\mathrm{eV}\right)$	$\sigma_{0P}\;\left(S/m\right)$	$H_{P}^{0}\;\left(\mathrm{eV}\right)$	α
Wadsleyite	399 (311)	1.49 (10)	7.74 (4.08)	0.68 (3)	0.02 (2)
Ringwoodite	838 (442)	1.36 (5)	27.7 (9.6)	1.12 (3)	0.67 (3)

**Table 3 sensors-23-01249-t003:** Values of parameters contained in Equation (14).

$\Delta H^{Vacancy}\;(\mathrm{KJ}/\mathrm{mol})$	$\log\sigma_{0}{}^{Vacancy}\;(S/m)$	$\Delta H^{Polaron}\;(\mathrm{KJ}/\mathrm{mol})$	$\log\sigma_{0}{}^{Polaron}\;(S/m)$	$\Delta H^{Hydrous}\;(\mathrm{KJ}/\mathrm{mol})$	$\log\sigma_{0}{}^{Hydrous}\;(S/m)$	*α*
239 ± 46	5.07 ± 1.32	144 ± 16	2.34 ± 0.67	89 ± 9	−1.37 ± 0.45	1.79 ± 0.55

## Data Availability

The data used for inversion can be downloaded from the World Data Center and be obtained from the Geomagnetic Network of China.

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
