# Peer review of "A Mantle Plume Beneath South China Revealed by Electrical Conductivity Obtained from Three-Dimensional Inversion of Geomagnetic Data"

_sensors, 2023, doi:10.3390/s23031249_

Round 1
Reviewer 1 Report
1. The inversion data does not seem to have quality analysis. In the inversion fitting results (Fig. 6), the error bar of the original data seems to be large, so please explain;
2. In the analysis of inversion results (Fig. 7), please explain why the author chose these depths for analysis, and what are the special reasons;
3. It is suggested that the author increase the analysis of Hainan mantle plume, which can be compared with other data to strengthen the support of this argument.
Author Response
Comments and Suggestions for Authors
- The inversion data does not seem to have quality analysis. In the inversion fitting results (Fig. 6), the error bar of the original data seems to be large, so please explain;
Response:
1) The variations on squared coherency of C-response, which are always treated as a quality indicator of C-responses, are drawn in Fig. 2, showing that most of the responses are with good quality, while some stations in poor quality.
2) The estimation of C-responses in frequency-domain strongly depends on the data length in time-domain. However, the data length varies at stations, besides the observed data and strongly affected by the environment noises and ocean effect, leading to a huge variability both in real and imaginary parts of C-responses, especially for responses in long-period, corresponding to the large error bar of C-responses. Considering the variations of C-responses, the L1-norm inversion, in which the data misfit is measured by L1-norm measurement, is used in this paper to suppress the influence of data with large noises.
The related description has been added into the manuscript.
- In the analysis of inversion results (Fig. 7), please explain why the author chose these depths for analysis, and what are the special reasons;
Response: Fig. 7 displays the electrical conductivity distribution in the mantle (250-1600 km) beneath South China, corresponding to the sensitive depth of GDS. The discussed features are mainly focused in 520-900 km, which is the most sensitive depth of GDS, making the conductive anomaly more reliable.
The related details are corrected in the manuscript, lines 254-255.
- It is suggested that the author increase the analysis of Hainan mantle plume, which can be compared with other data to strengthen the support of this argument.
Response: Thanks for the comments. More analysis and comparison have been added into the manuscript.

Reviewer 2 Report
This paper has reported a major conductor in an electrical conductivity model constrained by geomagnetic depth soundings (GDS) at 15 sites in the South China area. The authors interpreted this conductor as evidence of a mantle plume (the Hainan plume in the paper). GDS is one of the important methods to detect subsurface electrical conductivities at depths up to the earth's lower mantle. The structure of the earth's mantle is poorly understood, and the origination and structure of the mantle plume is a topic widely concerned in recent years. Therefore, the topic selection is interesting. However, it is not at all convinced that the current version of the paper is appropriate for publication. This paper has to be greatly improved in terms of the significance and reliability of the results and the rationality of the explanation. In the following, the main issues concerned are discussed, and then some personal suggestions are provided for the author's reference.
1. Involved inversion data had been analyzed in previously published works. Most of the data involved in this paper were, to my knowledge, interpreted by Yuan et al. (2020), and Yao et al. (2022) analyzed one of these data (site GZH). To improve the significance of this study, those and any other previous studies on these data are supposed to mention in the text. This would be beneficial to enhance the advancements of this new research and address the implications of this work. In addition, at site GZH the data curves show different amplitudes in real and imaginary components from Yao et al. (2022), a reference paper listed in the text. Please explain the causes of the differences.
Two papers mentioned above:
Yao, H., Ren, Z., Tang, J., and Zhang, K., 2022, A Multi‐Resolution Finite‐Element Approach for Global Electromagnetic Induction Modeling With Application to Southeast China Coastal Geomagnetic Observatory Studies: Journal of Geophysical Research: Solid Earth, v. 127, no. 8.
Yuan, Y., Uyeshima, M., Huang, Q., Tang, J., Li, Q., and Teng, Y., 2020, Continental-scale deep electrical resistivity structure beneath China: Tectonophysics, v. 790, p. 228559.
2. A number of geophysical models for this region have been revealed by seismic and gravity data. Some comparisons to other geophysical models of this study area are necessary. It would be nice to address how these models differ from the electrical conductivity model, what causes the differences, and the valuable information provided by different geophysical models. As for the seismic velocity model (Xia et al.,2016) mentioned in the text, it is better to plot the horizontal conductivity slices at the same depths as the velocity model provided by Xia et al.,2016. It would be convenient to see the differences between the two models by doing so. In Figure 7, please check if the areas enclosed by black lines correctly indicate the spatial range of low velocities at corresponding depths. The locations and range of the low velocities in the mantle are varied with depths in the paper by Xia et al., 2016. This feature of change is also shown in other published papers listed below:
He, C., and Santosh, M., 2021, Mantle Upwelling Beneath the Cathaysia Block, South China: Tectonics, v. 40, no. 4.
Huang, J., and Zhao, D., 2006, High-resolution mantle tomography of China and surrounding regions: Journal of Geophysical Research, v. 111, no. B9.
Li, C., and van der Hilst, R. D., 2010, Structure of the upper mantle and transition zone beneath Southeast Asia from traveltime tomography: Journal of Geophysical Research, v. 115, no. B7.
Zhao, D., Toyokuni, G., and Kurata, K., 2021, Deep mantle structure and origin of Cenozoic intraplate volcanoes in Indochina, Hainan and South China Sea: Geophysical Journal International, v. 225, no. 1, p. 572-588.
Toyokuni, G., Zhao, D., and Kurata, K., 2022, Whole‐Mantle Tomography of Southeast Asia: New Insight Into Plumes and Slabs: Journal of Geophysical Research: Solid Earth, v. 127, no. 11.
3. In section 4, more detailed processes and information on how the final temperatures came out are needed instead of listing reference papers only. Several variables are shown in the equations (12-14), so the specific parameters applied to calculate the required temperatures have to be provided in the text. Moreover, it would be nice to discuss multiple candidates to account for the significant conductors in the mantle revealed by the data. If water is not an explanation, as addressed in lines 285-286 by the authors, please provide evidence or literature.
Other suggestions:
4. Line 166: 'in small periods' means 'at short periods'?
5. Lines 212-215 describe the horizontal grids in two sizes used for the inversion model, but it is not easy to understand how these grids were constructed.
6. Line 229 indicates that the author had included unreliable data in the inversions. I recommend that the unreliable data should be deleted before running the inversions.
7. It would be great if the authors could provide some of the inversion results with different regularizations indicated in lines 224-226 and 231.
8. There are two conductors in Figure 10a. It is confusing which one is A. It is better to label conductor A in Figures 7 and 10, which was mentioned repeatedly in the text.
9. Line 245 indicates a region with a conductivity of 7 S/m at a depth of ~600 km. It is necessary to address the reason for this unusual conductivity value in the model.
10. To evaluate if features in the model are required by the data, it is necessary to show the new responses at all sites from the constrained inversion in Section 3.2, particularly the one on the top of conductor A. Or, it is better to explain why other responses are not shown.
Author Response
Comments and Suggestions for Authors
This paper has reported a major conductor in an electrical conductivity model constrained by geomagnetic depth soundings (GDS) at 15 sites in the South China area. The authors interpreted this conductor as evidence of a mantle plume (the Hainan plume in the paper). GDS is one of the important methods to detect subsurface electrical conductivities at depths up to the earth's lower mantle. The structure of the earth's mantle is poorly understood, and the origination and structure of the mantle plume is a topic widely concerned in recent years. Therefore, the topic selection is interesting. However, it is not at all convinced that the current version of the paper is appropriate for publication. This paper has to be greatly improved in terms of the significance and reliability of the results and the rationality of the explanation. In the following, the main issues concerned are discussed, and then some personal suggestions are provided for the author's reference.
- Involved inversion data had been analyzed in previously published works. Most of the data involved in this paper were, to my knowledge, interpreted by Yuan et al. (2020), and Yao et al. (2022) analyzed one of these data (site GZH). To improve the significance of this study, those and any other previous studies on these data are supposed to mention in the text. This would be beneficial to enhance the advancements of this new research and address the implications of this work. In addition, at site GZH the data curves show different amplitudes in real and imaginary components from Yao et al. (2022), a reference paper listed in the text. Please explain the causes of the differences.
Two papers mentioned above:
Yao, H., Ren, Z., Tang, J., and Zhang, K., 2022, A Multi‐Resolution Finite‐Element Approach for Global Electromagnetic Induction Modeling with Application to Southeast China Coastal Geomagnetic Observatory Studies: Journal of Geophysical Research: Solid Earth, v. 127, no. 8.
Yuan, Y., Uyeshima, M., Huang, Q., Tang, J., Li, Q., and Teng, Y., 2020, Continental-scale deep electrical resistivity structure beneath China: Tectonophysics, v. 790, p. 228559.
Response: The responses of GZH are discussed in the work of Yuan et al. (2020), and Yao et al. (2022). In Yuan et al. (2020), data at GZH fails to satisfy the 1-D structure assumption of the ρ+ test, which is applied to converted the data to 1-D mantle structure, and are not further discussed and shown in the paper. In Yao et al. (2022), the observed C-responses are estimated from the hourly mean value of geomagnetic fields following the method of Semenov and Kuvshinov (2012). In our paper, the C-responses are processed by the method that we modify from Chave and Thomson (2004). We speculate that the details in processing progress are different, especially the referenced site used for correlate the geomagnetic field. However, both in the inversion results of Yao et al. (2022) and our work, a high resistivity in the mantle transition zone is present beneath GZH, indicating the reliability of our result.
Mentioned reference:
Chave A D, Thomson D J. 2004, Bounded influence magnetotelluric response function estimation. Geophysical Journal International, v. 157, 988–1006.
- A number of geophysical models for this region have been revealed by seismic and gravity data. Some comparisons to other geophysical models of this study area are necessary. It would be nice to address how these models differ from the electrical conductivity model, what causes the differences, and the valuable information provided by different geophysical models. As for the seismic velocity model (Xia et al.,2016) mentioned in the text, it is better to plot the horizontal conductivity slices at the same depths as the velocity model provided by Xia et al.,2016. It would be convenient to see the differences between the two models by doing so. In Figure 7, please check if the areas enclosed by black lines correctly indicate the spatial range of low velocities at corresponding depths. The locations and range of the low velocities in the mantle are varied with depths in the paper by Xia et al., 2016. This feature of change is also shown in other published papers listed below:
He, C., and Santosh, M., 2021, Mantle Upwelling Beneath the Cathaysia Block, South China: Tectonics, v. 40, no. 4.
Huang, J., and Zhao, D., 2006, High-resolution mantle tomography of China and surrounding regions: Journal of Geophysical Research, v. 111, no. B9.
Li, C., and van der Hilst, R. D., 2010, Structure of the upper mantle and transition zone beneath Southeast Asia from traveltime tomography: Journal of Geophysical Research, v. 115, no. B7.
Zhao, D., Toyokuni, G., and Kurata, K., 2021, Deep mantle structure and origin of Cenozoic intraplate volcanoes in Indochina, Hainan and South China Sea: Geophysical Journal International, v. 225, no. 1, p. 572-588.
Toyokuni, G., Zhao, D., and Kurata, K., 2022, Whole‐Mantle Tomography of Southeast Asia: New Insight Into Plumes and Slabs: Journal of Geophysical Research: Solid Earth, v. 127, no. 11.
Response:
The depth of the background model used for our 3D inversion is not exactly same with that used in seismic velocity model (Xia et al.,2016). The difference makes it difficult to compare the structure of electrical and seismic in a certain depth. Therefore, in fig. 7, we drew the most adjacent slices provided by Xia et al. 2016. Considering that the electrical structure not covered by geomagnetic stations cannot be well constrained by GDS, we only showed the seismic structure in the region covered by our selected stations.
The inversion results of other geophysical models mentioned by the reviewer has been compared with the electrical structure obtained in this paper, the result shows a good agreement. Related discussion has been added in the manuscript.
Rewritten (lines: 371-379): Our deduction is also supported by previous seismological images. Xia et al. [10] discovered a continuous low-velocity anomaly in and around the MTZ, which extends down to the lower mantle deeper than 1100 km, as marked by the black circles in Fig. 7. Considering that the electrical structure not covered by geomagnetic stations cannot be well constrained by GDS, we only show the seismic structure in the region covered by our selected stations. The low-velocity anomaly is suggested to be the tail and head of the Hainan plume, which originates in the lower mantle and feeds the Hainan hotspot. Other wave velocity structures [49–51] and perturbations of 410 km and 660 km discontinuity [41] also suggest the existence of the Hainan plume.
- In section 4, more detailed processes and information on how the final temperatures came out are needed instead of listing reference papers only. Several variables are shown in the equations (12-14), so the specific parameters applied to calculate the required temperatures have to be provided in the text. Moreover, it would be nice to discuss multiple candidates to account for the significant conductors in the mantle revealed by the data. If water is not an explanation, as addressed in lines 285-286 by the authors, please provide evidence or literature.
Response: The value of parameters has been listed in the text and the table 2 & 3. The exclusion of water is explained more detailed in the text.
Rewritten (lines: 303-307): The contribution of water can be excluded because there is little evidence of a slab that penetrates or stagnates in the MTZ beneath South China. This exclusion can be strengthened by the elevated 660 km discontinuity at the location measured by the receiver function technique [41], which is contradictory to the depression caused by water [42].
Other suggestions:
- Line 166: 'in small periods' means 'at short periods'?
Response: Corrected.
- Lines 212-215 describe the horizontal grids in two sizes used for the inversion model, but it is not easy to understand how these grids were constructed.
Response: The set of horizontal grids used in inversion has been described more precisely.
Rewritten (lines: 223-226): To eliminate the OIEs, a lateral grid of 1°´1° of the surface layer with a thickness of 12.65 km is considered to more precisely describe the distribution of sea and continent to generate the shell in 3°´3° for calculation and ensure sufficiently accurate accounting for OIEs [28].
- Line 229 indicates that the author had included unreliable data in the inversions. I recommend that the unreliable data should be deleted before running the inversions.
Response: C-Responses with relatively large error bars are considered to be unreliable data in our papers. In our previous work (Li et al., 2020), we have proved that L1-norm inversion, in which the data misfit is measured by L1-norm measurement instead of L2-norm in traditional inversion, can suppress the influence caused by unreliable data and obtain a reliable inversion result. Therefore, L1-norm is used in our manuscript and the inversion result is believed to be reliable.
Paper mentioned above:
Li, S.; Weng, A.; Zhang, Y.; Schultz, A.; Li, Y.; Tang, Y.; Zou, Z.; Zhou, Z. Evidence of Bermuda Hot and Wet Upwelling from Novel Three-Dimensional Global Mantle Electrical Conductivity Image. Geochemistry, Geophys. Geosystems 2020, 21, doi:10.1029/2020GC009016.
- It would be great if the authors could provide some of the inversion results with different regularizations indicated in lines 224-226 and 231.
Response: The results of inversions with initial regularization parameter equaling 100 and stations excluding responses with large RMS are shown in the adding Fig. 6 to strengthen the reliability of our preferred model. Inversion results share a similar conductive feature beneath South China in the deep upper mantle, mantle transition zone, and lower mantle.
- There are two conductors in Figure 10a. It is confusing which one is A. It is better to label conductor A in Figures 7 and 10, which was mentioned repeatedly in the text.
Response: Thanks for the comments. Both the two conductors are anomaly A, and they are the portion of anomaly A in different depths. We label A in Figures 7 and 10 (in the revised manuscript is Figures 8 and 11) as commented by the reviewer to make the definition of anomaly A clearer.
- Line 245 indicates a region with a conductivity of 7 S/m at a depth of ~600 km. It is necessary to address the reason for this unusual conductivity value in the model.
Response: The high conductivity is mainly caused by the peculiarity of electromagnetic induction. We explain the reason in the following text.
Rewritten (lines: 258-262): The strongest variation in conductivity occurs in the lower MTZ, with a conductivity reaching approximately 7 S/m at the centre. However, this extremely high conductivity is not suitable for further analysis, because the electromagnetic induction is primarily sensitive to the integrated conductance of a conductivity body which prefers to generate a compensating higher conductivity in the core [37].
- To evaluate if features in the model are required by the data, it is necessary to show the new responses at all sites from the constrained inversion in Section 3.2, particularly the one on the top of conductor A. Or, it is better to explain why other responses are not shown.
Response: The most obviously variations on C-response of a conductor is located in the edge of the conductor instead of the center, which we have present in the other paper we have submitted but not published. The following figures are the C-response at different periods of a conductive mantle plume, and it can illustrate the variations of C-responses.
Fig. R1. Electrical conductivity calculated from the temperature (left panel) and C-response variations (right panel) of the mantle plume R1b (differences of responses between the plume R1b and background model at different periods, calculated by ). Dashed circles mark the outermost edge of plume R1b whose center is at (180°E, 40°N). The upper and lower panel show the variations of real and imaginary components of C-responses, respectively.
